# Evaluation of a commercial AI-assisted cell counting software for dopaminergic neurons across species

Ken Kunugitani[1,2], Masanori Sawamura[3*], Tomoyuki Taguchi[3], Tetsuya Hirato[3], Norihito Uemura[3,4], Takashi Ayaki[3], Etsuro Nakanishi[3], Hodaka Yamakado[5], Tomoyuki Ishimoto[3], Hirotaka Onoe[6], Tadashi Isa[7], Riki Matsumoto[3], Ryosuke Takahashi[3]

1 Department of Pharmacology, Kyoto University Graduate School of Medicine, Kyoto, Japan, 2 Kyoto University Hospital Integrated Clinical Education Center, Kyoto, Japan, 3 Department of Neurology, Kyoto University Graduate School of Medicine, Kyoto, Japan, 4 Department of Neurological Disease Control, Osaka Metropolitan University Graduate School of Medicine, Osaka, Japan, 5 Department of therapeutics for Multiple System Atrophy, Kyoto University Graduate School of Medicine, Kyoto, Japan, 6 Human Brain Research Center, Kyoto University Graduate School of Medicine, Kyoto, Japan, 7 Department of Physiology and Neurobiology, Kyoto University Graduate School of Medicine, Kyoto, Japan

* masawa@kuhp.kyoto-u.ac.jp

## Abstract

Quantification of dopaminergic neurons in the substantia nigra pars compacta (SNc) of animal models is important for understanding the pathogenesis of Parkinson's disease (PD). However, conventional manual cell counting method requires the time and effort, and has limited reproducibility due to inter- and intra-examiner variability. Here, we demonstrate that a commercially available convolutional neural network–based artificial intelligence (AI) counting method (TruAI, OLYMPUS, Tokyo, Japan) enables robust and reproducible quantification of TH-positive dopaminergic neurons in mouse, marmoset, and human SNc samples when compared with conventional manual counting. AI-based counting showed a strong correlation with manual counting across mouse, marmoset, and human samples. Good agreement between AI-based and manual counting was observed in mouse and marmoset samples, supporting the applicability of this approach for cross-species quantification of dopaminergic neurons. In the mouse model treated with α-syn preformed fibrils (PFFs), AI-based counting detected a significant reduction in TH-positive neurons consistent with expert manual counting. Non-experts exhibited greater intra-examiner variability than an expert, indicating that the reliability of manual counting depends on experience. Overall, AI-based quantification provides a robust and objective approach for TH-positive cell counting and may improve reproducibility in dopaminergic neuron analysis, particularly for non-expert users and cross-species studies of PD.

**Data availability statement:** All relevant data are within the manuscript and its Supporting information files.

**Funding:** The author(s) received no specific funding for this work.

**Competing interests:** The authors have declared that no competing interests exist.

## Introduction

Parkinson's disease (PD) is a neurodegenerative disorder characterized by a decrease in dopaminergic neurons in the substantia nigra pars compacta (SNc) and pathological accumulation of α-synuclein (α-syn) [1]. The detailed mechanisms of PD pathogenesis remain unclear; thus, for a better understanding of the condition, animal models of PD have been actively studied [2–4]. Since the loss of dopaminergic neurons in the SNc is a major pathophysiology of PD [1], accurate enumeration of dopaminergic neurons is essential for the development of models and therapeutics for PD pathology. Tyrosine hydroxylase (TH) is an enzyme that synthesizes the precursor of dopamine [5,6], L-3,4-dihydroxyphenylalanine, and quantification of TH-positive cells in the SNc is considered the gold standard for evaluating dopaminergic neurons in rodent models of PD [7]. However, it is difficult to automate the enumeration of TH-positive cells using classical image processing because of their irregular size and shape, unclear boundaries, and high cell density; thus, most studies on PD quantify TH-positive cells manually [8–10]. However, cell counting performed by human examiners limits the objectivity and reproducibility of results and imposes a burden on the experimenter.

In recent years, the development of machine learning technology in the field of artificial intelligence (AI) has led to the automation of tasks traditionally performed by humans [11]. The same is true for biological experiments. Recently, computerized target detection algorithms [12–17] and image analysis tools [18–20] have emerged, offering advantages such as automated measurements, elimination of experimenter bias, data reproducibility, elimination of human error, time and human resource savings, and high-speed high-volume analyses. In particular, deep learning methods using convolutional neural networks (CNNs) have attracted considerable attention. By learning features directly from a set of data, it is now possible to make measurements that were previously difficult with classical methods [21,22]. To our knowledge, multiple CNN-based AIs have been previously reported for enumerating TH-positive cells in the SNc of neurotoxic PD mouse models [23–27]; however, they are yet to be recognized as the gold standard. Moreover, while non-human primate PD models have been reported [3,28], no method has been established for automatically counting TH-positive cells in non-human primates. Here we investigated whether a commercially available CNN-based AI (TruAI, OLYMPUS, Tokyo, Japan) could serve as an alternative to conventional manual counting for enumerating TH-positive cells in mouse and marmoset PD models treated with α-syn preformed fibrils (PFFs), as well as in human PD patients.

## Results

### Phosphorylated-α-synuclein (p-α-syn) staining confirmed PFFs-treated mice as Lewy body diseases (LBD) model

As previously reported [29], phosphate-buffered saline (PBS) or PFFs were administered to the unilateral striatum of bacterial artificial chromosome (BAC) transgenic mice expressing familial PD-linked A53T mutant human α-syn (A53T BAC-*SNCA*

transgenic mice), and dopaminergic neurons in the ipsilateral substantia nigra were evaluated with immunohistochemistry using anti p-α-syn. Compared with the PBS group (Fig 1a, b), the PFFs group exhibited p-α-syn aggregates in the SNc (Fig 1c, d).

## Validation of AI-based counting against manual counting in mouse SNcd samples

For mouse and marmoset sample counting, SNc was divided into four segments according to the morphology, size, and distribution of the cell as per a previous study [30]: dorsal (SNcd), medial (SNcm), ventral (SNcv), and lateral (SNcl). In this study we focused on the SNcd, where neuronal loss is reported to be the most severe in PD [10].

To evaluate the applicability of AI-based counting for TH-positive cell quantification, we compared its performance with conventional manual counting by an expert with more than five years of experience (Fig 1e–h). AI-based counting showed a very strong correlation with expert manual counting (r = 0.997, p < 0.001, Pearson correlation, Fig 1i), supporting the feasibility of AI-based quantification in mouse SNcd samples. Bland–Altman analysis further demonstrated good agreement between AI-based and manual counting, revealing a small systematic bias in which AI-based counting slightly overestimated TH-positive cell numbers compared with manual counting, with narrow limits of agreement (−29.4 to −1.1 cells; Fig 1j). Importantly, no trend toward increased differences at higher or lower cell counts was observed, indicating the absence of proportional bias. Together, these results support the quantitative reliability of AI-based counting in mouse SNcd samples. Consistent with manual counting, AI-based counting successfully identified a significant reduction in TH-positive cells in the PFFs group (p < 0.001, Welch's *t*-tes*t*, Fig 1k), demonstrating that AI-based quantification preserves biologically meaningful group differences. The performance metrics yielded a precision of 81.4%, a recall of 88.2%, and an F1-score of 84.7% (Table 1). S1 Table in S1 File provides raw counting data, including true positive (TP), false positive (FP), and false negative (FN) values based on manual counting.

Additionally, the accuracy of AI-based counting was influenced by staining quality. When a different TH antibody (Millipore, #AB152, rabbit, 1:800 dilution) was used [10], the lower staining intensity increased false negatives (Fig 1l, m), highlighting the importance of contrast in AI-based quantification.

## Manual counting showed intra-examiner variability over time in repeated counting experiment for selected mouse samples

To examine over time variability of AI-based and manual counting, we performed repeated counting experiment using one mouse each from the PBS and the PFFs groups. AI-based counting showed consistent results over time, whereas manual counting exhibited both inter- and intra-examiner variability. Non-experts (less than 5 years of manual counting experience) exhibited greater intra-examiner variability than an expert, as indicated by the higher coefficients of variation (CV) in Table 2. These results suggest that, within this limited repeated-counting experiment, the variability of manual counting can be influenced by the examiner's experience.

## Validation of AI-based counting against manual counting in marmoset SNcd samples

Next, we revealed that our neural network is applicable for another PD model, using PFFs-treated marmoset samples from the previous report [3]. Since there is no expert in counting marmoset TH-positive cells, three non-experts (examiners 1–3) independently counted TH-positive cells and compared the results with AI-based counting. AI-based counting could identify TH-positive cells comparable to manual counting (Fig 2a–2d). AI-based counting showed strong correlations with each of manual counting (r = 0.96, 0.95, 0.95; p < 0.001, Pearson correlation; Fig 2e–g), with performance metrics of 91.3% precision, 87.8% recall, and 89.5% F1-score (Table 1). Bland–Altman analysis demonstrated good agreement between AI-based counting and the averaged manual counts obtained from three independent examiners, with a small systematic bias and narrow limits of agreement (−23.8 to +8.8 cells; Fig 2h). Differences between the two methods were largely consistent across the range of cell counts, although a minor trend toward more negative differences at higher

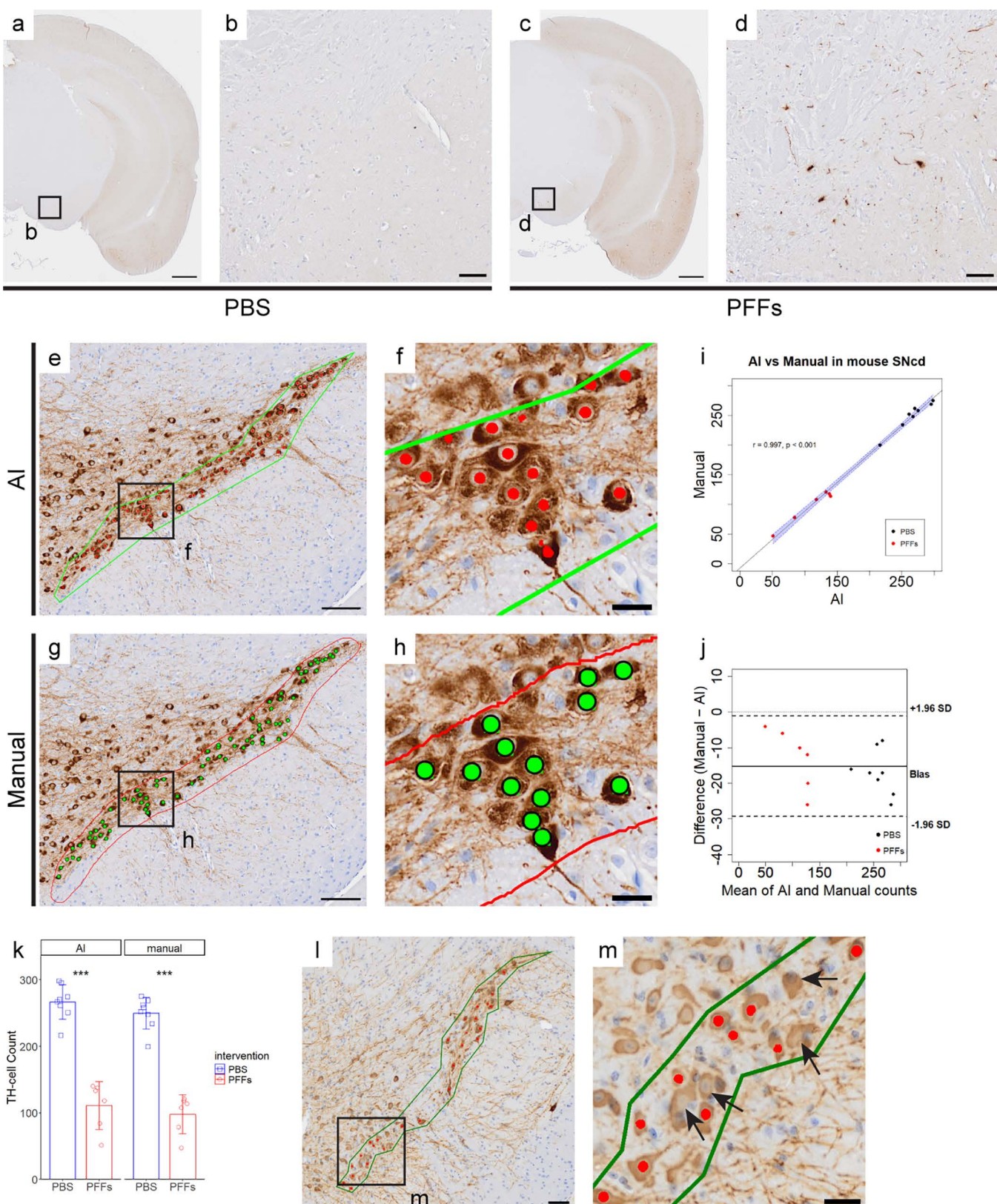

**Fig 1. AI-based counting detects mouse TH-positive cells comparable to expert's manual counting.** (a-d) Representative immunohistochemical staining images for p-α-syn in PBS (a and b) and PFFs (c and d)-injected heterozygous A53T BAC-*SNCA* transgenic mice. Low-power images (a and c)

and high-power images focused on the SNc (b and d). Scale bar; a and c, 500 μm; b and d, 100 μm. (e-h) Representative images of AI-based counting (e and f) and manual counting (g and h) in mouse SNcd. Scale bar; e and g, 100 μm; f and h, 20 μm. (i) AI-based counting shows a very strong correlation with manual counting (r = 0.997, ***p < 0.001, Pearson correlation). Each data point represents the total number of TH-positive cells per individual animal, obtained by summing counts across all analyzed sections for that animal. The x-axis shows the AI-based counting results, and the y-axis shows the corresponding manual counting results. Black dots represent PBS-injected mice, red dots represent PFFs-injected mice and the blue shaded area indicates the 95% confidence interval. (j) Bland–Altman plot comparing AI-based and manual TH-positive cell counts in mouse SNcd samples. Differences (Manual−AI) of TH-positive cell counts per animal are plotted against the mean of the two methods. The solid line represents the mean difference (bias), and dashed lines represent the 95% limits of agreement (±1.96 SD). Points are color-coded by intervention (PBS and PFFs). (k) Both AI-based and manual counting detect significant decrease in TH-positive cells in the PFFs group compared to the PBS group. Mean TH-positive cell counts per animal obtained by AI-based counting are 267 ± 26 for PBS mice and 111 ± 36 for PFFs mice, while those obtained by manual counting are 250 ± 24 for PBS mice and 98 ± 29 for PFFs mice. Each data point represents the total TH-positive cell count per individual animal, with bars indicating the mean and error bars representing the standard deviation (SD). ***p < 0.001, Welch's *t*-test. (l and m) Specimens with weak staining show increased false negatives (m, arrows). Scale bar; l, 50 μm; m, 20 μm.

**Table 1. Formulas and results for precision, recall, and F1-score of AI-based counting versus manual counting.**

| Metrics | Mouse Score | Marmoset Score | Human Score |
|---|---|---|---|
| Precision = TP/(TP + FP) | 81.4% | 91.3% | 83.0% |
| Recall = TP/(TP + FN) | 88.2% | 87.8% | 68.5% |
| F1-score = 2*Precision*Recall/(Precision + Recall) | 84.7% | 89.5% | 75.0% |

TP, true positive; FP, false positive; FN, false negative.

**Table 2. Results of repeated counting at one-week intervals using images from one representative mouse each from the PBS and PFFs groups. The coefficients of variation (CV) highlight the intra-examiner variability for each tester.**

| Tester | Intervention | 1st trial | 2nd trial | 3rd trial | Mean | SD | CV (%) |
|---|---|---|---|---|---|---|---|
| AI | PBS | 261 | 261 | 261 | 261 | 0 | 0 |
| | PFFs | 118 | 118 | 118 | 118 | 0 | 0 |
| Non-expert 1 | PBS | 250 | 249 | 256 | 252 | 3.1 | 1.2 |
| | PFFs | 102 | 99 | 109 | 103 | 4.2 | 4.1 |
| Non-expert 2 | PBS | 232 | 234 | 250 | 239 | 8.1 | 3.4 |
| | PFFs | 96 | 94 | 98 | 96 | 1.6 | 1.7 |
| Expert | PBS | 248 | 246 | 246 | 247 | 0.9 | 0.4 |
| | PFFs | 91 | 92 | 91 | 91 | 0.5 | 0.5 |

SD, standard deviation; CV, coefficient of variation; PBS, phosphate-buffered saline; PFFs, preformed fibrils.

values was noted. These results further support the quantitative reliability of AI-based counting for TH-positive cell quantification in marmoset samples. The original data are presented in S2 Table in S1 File.

When we counted TH-positive cells in the marmoset samples using the mouse neural network, the results showed an increase of false negatives (S1a Fig in S1 File). These results indicate that species-specific neural network models are required to achieve optimal detection performance across different species.

## Applicability of AI-based counting to human SNc samples

We further examined whether our AI-based counting method is applicable for quantifying human dopaminergic neurons by analyzing human PD patient samples and age-matched control samples from our previous report [31]. Representative images of AI-based counting could identify TH-positive cells comparable to manual counting (Fig 3a–d). AI-based

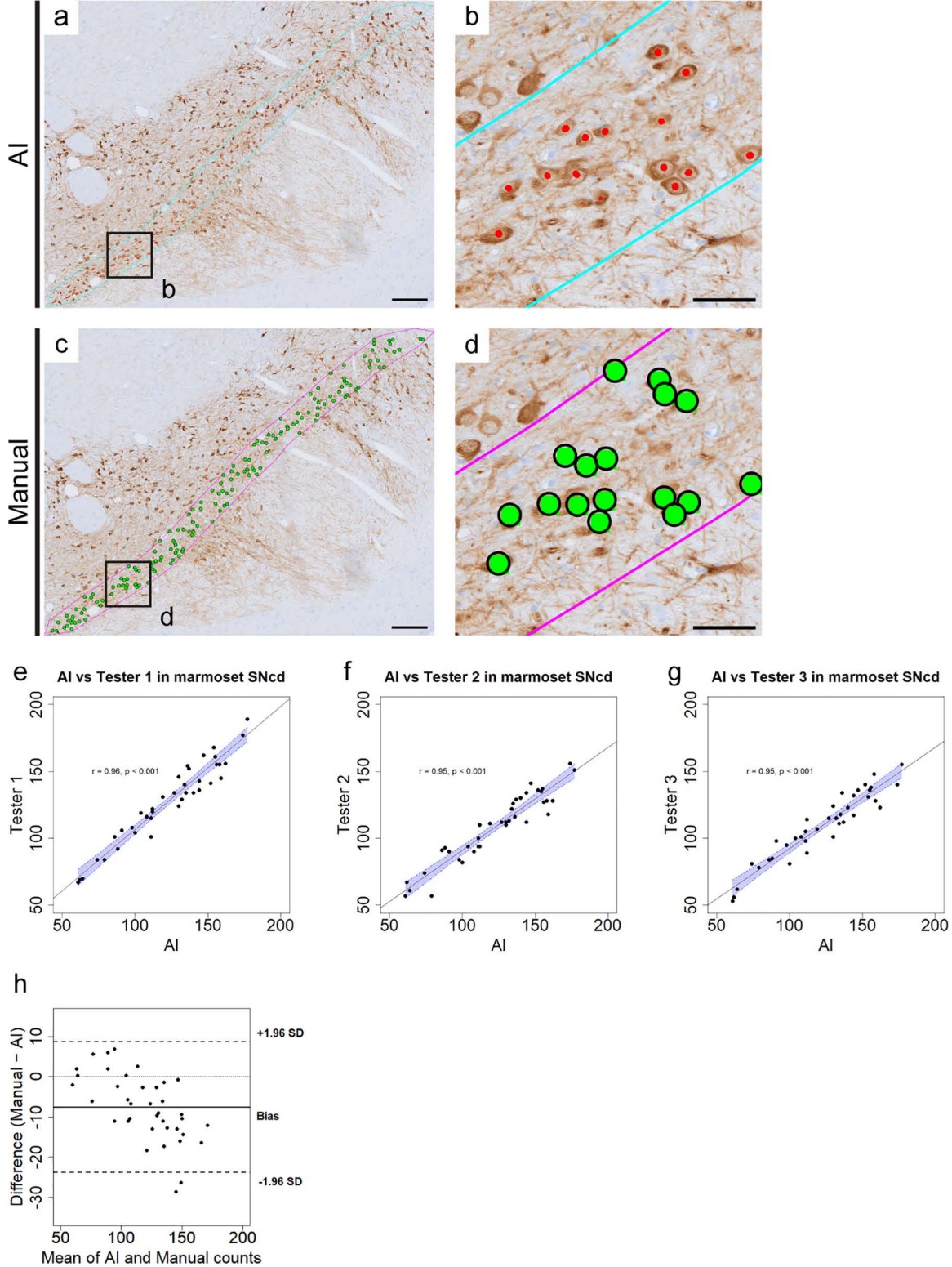

**Fig 2. AI-based counting is also applicable for marmoset TH-positive cells.** (a-d) Representative images of AI-based counting (a and b) and manual counting (c and d) results in marmoset SNcd. Scale bar; a and c, 200 μm; b and d, 50 μm. (e-g) AI-based counting shows strong correlations with each of the three manual counting (r = 0.96, 0.95, 0.95; ***p < 0.001, Pearson correlation). Each data point represents the TH-positive cell count for an

individual unilateral SNc obtained from a single analyzed section. The x-axis shows the AI-based counting results, and the y-axis shows the corresponding manual counting results. The blue shaded area indicates 95% confidence interval. (h) Bland–Altman plot comparing AI-based and manual TH-positive cell counts in marmoset SNcd samples. Differences (Manual − AI) between AI-based counts and the averaged manual counts for an individual unilateral SNc from three independent examiners are plotted against their mean. The solid line indicates the mean difference (bias), and dashed lines indicate the 95% limits of agreement (±1.96 SD).

counting showed strong correlations with manual counting performed by three independent examiners (r = 0.96, 0.97, 0.98; p < 0.001, Pearson correlation; Fig 3e–g). Given the limited number of available human sections and the absence of an established reference standard, agreement analysis was not used as a primary quantitative validation for human samples. Performance metrics from human samples are summarized in Table 1 and indicate moderate detection performance under the tested conditions. Consistent with manual counting, AI-based counting detected a trend toward a decrease in TH-positive cell density in PD patient samples compared to controls; however, the difference did not reach statistical significance (n.s., Welch's t-test; Fig 3h). The analysis was based on a limited number of sections (three sections from 3 PD patients and six sections from 5 control subjects). The original data are presented in S3 Table in S1 File.

## Discussion

Here, we demonstrated that our AI-based counting method enables robust and reproducible quantification of TH-positive cells in comparison with conventional manual counting across mouse, marmoset, and human samples, within the scope of the datasets analyzed. Notably, while no established method exists for automatically counting TH-positive cells in non-human primates, our study provides the first demonstration of the applicability of AI-based counting to a marmoset PD model, supported by independent manual counting by multiple examiners. These results underscore the robustness of the method in a biologically and technically challenging primate model. In addition, repeated counting experiments indicated that AI-based counting yielded stable results over time, whereas manual counting exhibited examiner-dependent variability under the tested conditions. These findings suggest that AI-based approaches may contribute to improved objectivity and reproducibility in dopaminergic neuron quantification.

To our knowledge, multiple deep-learning–based approaches have been reported for automated quantification of TH-positive dopaminergic neurons in the mouse SNc [23–27]. The performance metrics of these models are summarized in Table 3. One of these models exhibited 69.0% precision, 79.3% recall, and 73.0% F1-score, which were slightly lower than those of the present study [25]. The remaining models showed high performance, with precision ranging from 86.0–95.3%, recall from 87.8–95.5%, and F1-scores from 88.0–95.3% [23,24,26,27]. These results indicate that CNN-based AI enables automated TH-positive cell counting, which is difficult to achieve using classical image-processing approaches. Most previously reported models were primarily evaluated using neurotoxic PD models, such as 6-hydroxydopamine (6-OHDA) [23] or 1-methyl-4-phenyl-1,2,3,6-tetrahydropyridine (MPTP) models [24,25], which are known to induce large, dose-dependent reductions in TH-positive neurons (approximately 50–90%) [32,33]. More recently, Chen et al. demonstrated a deep-learning–based quantification pipeline validated in both neurotoxic (MPTP) and PFF models [27], indicating that CNN-based approaches can be applied across distinct PD paradigms. In addition, Haghighi et al. also reported a CNN-based quantification framework applied to PD model mice [26], although the specific PD induction method was not explicitly categorized. Notably, α-synuclein PFF models are generally characterized by more gradual and modest reductions in TH-positive neurons compared with neurotoxic models. The present study focused on the α-syn PFFs model, which has been reported to show an approximately 38% decrease in TH-positive cells at 2 months post-injection [34]. Our results demonstrate that the proposed AI-based approach can reliably quantify TH-positive neurons even under conditions of relatively small neuronal loss. Taken together, these findings indicate that the method implemented in this study achieves performance comparable to previously reported CNN-based approaches while being applicable to PD models with varying degrees of neuronal loss. In addition, by leveraging a commercially available, user-friendly AI platform,

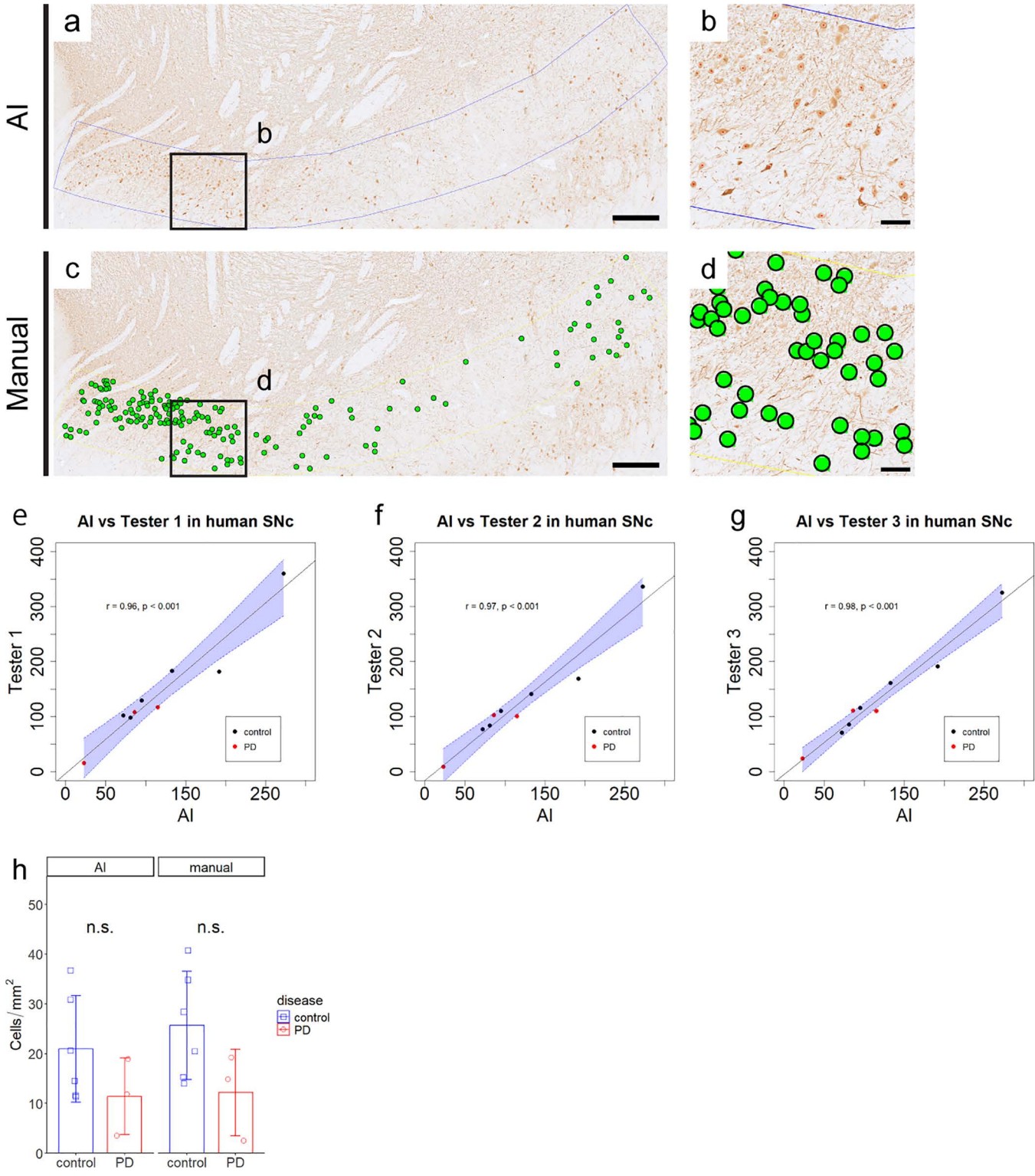

**Fig 3. AI-based counting is further applicable for human TH-positive cells.** (a-d) Representative image of AI-based counting (a and b) and manual counting (c and d) results in the human SNc. Scale bar; a and c, 500 μm, b and d, 100 μm. (e-g) AI-based counting shows a strong correlation with each of the three manual counting (r = 0.96, 0.97, 0.98; ***p < 0.001, Pearson correlation). Each data point represents the TH-positive cell count for an

individual unilateral SNc from a single analyzed section. The x-axis and y-axis show the AI-based and corresponding manual counting results, respectively. Black dots indicate control sections, red dots indicate PD patient sections, and the blue shaded area represents the 95% confidence interval. (h) Both AI-based and manual counting detect a trend toward a decrease in TH-positive cell density in PD patient samples compared to controls, though the difference is not statistically significant. The mean TH-positive cell density in the SNc (cells/mm$^2$) per section obtained by AI-based counting is $20 \pm 11$ for controls and $11 \pm 8$ for PD patients, while that obtained by manual counting is $26 \pm 11$ for controls and $12 \pm 9$ for PD patients. Each data point represents the TH-positive cell density measured in an individual unilateral SNc from a single section, with bars representing the mean and error bars representing the SD. n.s., Welch's *t-t*est.

**Table 3. Performance metrics of CNN-based AIs for counting TH-positive cells reported to date.**

| Report | Precision | Recall | F1-score |
|---|---|---|---|
| Penttinen AM et al., 2018 | 88.5% | 87.8% | 88.2% |
| Zhao S et al., 2018 | 95.0% | 91.9% | 93.4% |
| Kim D et al., 2023 | 69.0% | 79.3% | 73.0% |
| Haghighi et al., 2023 | 95.3% | 95.5% | 95.3% |
| Chen et al., 2024 | 86.0% | 91.0% | 88.0% |

our approach facilitates efficient training and application of deep-learning–based neuron quantification without requiring specialized programming expertise, thereby lowering the technical barrier for routine use in experimental neuroscience and histopathological analysis. Recent advances in generalizable segmentation frameworks, such as Cellpose [35], have improved the accuracy and robustness of automated cell detection across diverse imaging conditions. Integration of such advanced segmentation approaches may further enhance the accuracy and scalability of AI-based quantification of TH-positive neurons in future applications.

Optimized immunohistochemical processes remain critical for reliable AI-based quantification, as staining intensity directly affects image contrast during whole-slide processing and leads to variability in AI counting results, even though the training dataset incorporated reference images spanning a wide range of staining intensities. The main factors that decrease the signal-to-noise ratio (S/N ratio) of a specimen are antibody and diaminobenzidine (DAB) staining. We used two different primary antibodies for TH staining of specimens [10], however one showed a weaker staining and a lower S/N ratio than the other, making accurate measurement difficult. Therefore, it is important to compare multiple antibodies, select the antibody with the highest S/N ratio. Although DAB staining [36,37] is considered sensitive [38], it should be kept in mind that the staining intensity is strongly influenced by horseradish peroxidase (HRP) enzyme activity [39,40], reaction time, temperature, and HRP substrate concentration [41].

In conclusion, our AI-based counting method provides a reliable and objective approach for quantifying TH-positive cells in PD model animals. While further validation using larger and more diverse human datasets will be required, this method represents a practical alternative to conventional manual counting and has the potential to facilitate quantitative and reproducible analyses of dopaminergic neuron populations in experimental studies of PD.

## Limitations of the study

A limitation of the present study is the restricted number of human sections available for analysis, which resulted in limited sampling coverage along the rostro–caudal axis of the SNc. This constraint likely reduced the statistical power to detect neuronal loss in PD samples, despite the well-established pathology. Future studies using larger cohorts and anatomically systematic sampling will be required to fully validate the applicability of AI-based counting in human PD tissue.

Another limitation relates to the performance of TruAI in regions with extreme cell density or severe overlap of TH-positive neurons. TruAI relies on pattern recognition learned from reference images rather than explicit rule-based separation of densely packed cells; therefore, detection accuracy in such regions depends on whether similar patterns were

sufficiently represented in the training dataset. In the present study, no regions exhibiting extreme cellular crowding that would compromise reliable cell identification were observed. Nevertheless, under conditions of unusually high cell density, detection accuracy may be reduced, and careful visual validation remains important.

## Materials and methods

### Ethical approval

All experimental procedures were performed in accordance with the national guidelines and the ARRIVE (Animal Research: Reporting of *In Vivo* Experiments) guidelines. This study was approved by the Animal Research Committee of Kyoto University (mouse: MedKyo 24322; marmoset: MedKyo 16651) and the local ethics committee (human: R1038).

### Mouse samples

Seventeen heterozygous A53T BAC-*SNCA* transgenic mice [10] were used in this study (9 PBS-treated and 8 PFF-treated mice). To confirm the genotype of the A53T BAC-*SNCA* transgenic mice, polymerase chain reaction was performed as previously described [10]. From these animals, a total of 130 sections were obtained. Of these sections, 4 sections from 1 PBS-treated and 6 sections from 2 PFF-treated mice were used as reference images, and the remaining 72 sections from 8 PBS-treated and 48 sections from 6 PFF-treated mice were used for cell counting. The ipsilateral SNcd corresponding to the PBS- or PFF-treated side was analyzed in each section.

### Preparation of recombinant mouse α-syn monomers and PFFs

Mouse α-syn monomers and PFFs were synthesized as the previous reports [10,34,42–44], with minor modifications. Briefly, α-syn was expressed in *Escherichia coli* BL21 (DE3) (BioDynamics Laboratory) and purified by boiling and subsequent ion exchange chromatography using a Q Sepharose Fast Flow (GE Healthcare) system. After dialyzed against a dialysis buffer (150 mM KCl, 50 mM Tris-HCl, pH 7.5), the α-syn solution (5 mg/ml) was agitated at 37°C at 1000 rpm for 7 days and stored at −80°C. PFF was thawed and sonicated for 60 min (30 sec on/30 sec off) before injection.

As previously reported [34], 3-month-old mice were anesthetized with isoflurane and then stereotaxically inoculated with 2 μl of α-syn PFFs or PBS into the unilateral dorsal striatum (coordinates: Anterior-Posterior +0.2 mm relative to the bregma, Medial-Lateral +2.0 mm from the midline, Dorsal-Ventral −2.6 mm from the skull surface) using a 33-gauge Neuros syringe (Hamilton company).

### Immunohistochemistry for mouse samples

Two months after the injection of PFFs or PBS, mice were anesthetized with sevoflurane and transcardially perfused with cold PBS followed by 4% (w/v) paraformaldehyde (PFA) in PBS at 4°C overnight. Paraffinized brains were sectioned coronally in 8 μm thick on a microtome (EG1150, Leica Microsystems). For TH-positive cell counting, the entire SNc was sampled every 10th coronal section (80 μm apart) across its full rostro–caudal extent.

Immunohistochemical staining was performed as described previously [10,34], with minor modifications. The following primary antibodies were used; anti-phosphorylated-α-syn (p-α-syn) (Abcam, #ab51253, Rabbit, 1:20,000 dilution) and anti-TH (Millipore, #MAB318, Mouse, 1:5,000 dilution). The sections were incubated at 4°C with the primary antibodies for 2 days, following which appropriate secondary antibodies were added for p-α-syn (Nichirei Biosciences, # 414341) and TH (Nichirei Biosciences, #414322). The samples were processed for visualization using a peroxidase stain DAB kit (Nacalai tesque).

### Marmoset samples

Twenty-four sections from a two-year-old female marmoset (I5937; born at CLEA Japan, Inc.) that had received unilateral olfactory bulb injections of α-Syn PFFs were re-used from our previous study [10]. Of these sections, 5 were used as

reference images, and the remaining 19 sections were used for cell counting. The bilateral SNcd were analyzed in each section.

## Human samples

Human samples were re-used from our previous study [31] and consisted of 4 sections from 4 patients with PD and 11 sections from 5 age-matched control subjects. Of these sections, 1 section from 1 PD patient and 5 sections from 3 controls were used as reference images, and the remaining 3 sections from 3 PD patients and 6 sections from 5 controls were used for cell counting. The unilateral SNc was analyzed in each section.

## Image acquisition of mouse, marmoset, and human samples

Whole-slide images of TH-immunostained coronal sections were acquired using a slide scanner (SLIDEVIEW VS200, OLYMPUS, Tokyo, Japan).

## AI-based counting of TH-positive cells in mouse, marmoset, and human samples

AI-based enumeration of TH-positive cells was performed using a commercially available CNN-based platform (TruAI, OLYMPUS, Tokyo, Japan). The SNcd in mouse and marmoset samples, or the SNc of human samples, was defined as the region of interest (ROI). ROI segmentation was manually performed by an expert for mouse samples and by three non-experts for marmoset and human.

For reference images, SNcd of mouse and marmoset, or SNc of human were labeled as the background (in green), including regions outside SNcd or SNc that lacked TH-positive cells. Intact TH-positive cells with visible nuclei in SNcd or SNc were labeled as positive data (in red). In total, 1,353 mouse, 1,313 marmoset, and 1,398 human dopaminergic neurons were annotated and incorporated into the training datasets.

Reference images were intentionally selected to cover a wide range of TH staining intensities, from weak to strong signals, to ensure robust performance under realistic histopathological conditions. Representative low-intensity regions of the reference images are shown in S1b-d Fig in S1 File. In addition, images from PFF-treated mice and marmosets and PD patients were included to allow the model to learn morphological variations and distortions of TH-positive neurons associated with pathological conditions. These reference images were used exclusively for neural network training and were not included in the cell counting analysis to ensure an unbiased assessment of neural network performance and to prevent data leakage [45].

Neural networks were trained independently for each species for 25,000 iterations. The resulting similarity scores were 0.71 for mouse, 0.70 for marmoset, and 0.56 for human datasets. Following training, TH-positive cells within each ROI were automatically counted using the trained neural networks. Detailed parameter settings used for network training are summarized in the Supplementary Methods. The trained TruAI neural network files are provided as supplementary datasets entitled "neuralnetwork_THcell_mouse.nn," "neuralnetwork_THcell_marmoset.nn," and "neuralnetwork_THcell_human.nn." The workflows for AI-based and manual counting are shown in S2a Fig in S1 File (mouse), S3 Fig in S1 File (marmoset), and S4 Fig in S1 File (human).

## Manual counting of TH-positive cells in mouse, marmoset, and human samples

Intact TH-positive cells with visible nuclei in each ROI were counted using the multipoint tool in Fiji (ImageJ, Version 1.54j, National Institutes of Health, USA) [46]. Manual counting was performed by an expert (T.T.) for mouse samples and by three non-experts (K.K., M.S., and T.H.) for marmoset and human samples, and compared with AI counting. The expert tester (T.T.) is a board-certified neurologist specializing in Parkinson's disease, with 13 years of clinical experience and over eight years of research experience in Parkinson's disease model mice, including neuropathological evaluation and accurate quantification of dopaminergic neuronal loss [10].

### Repeated counting experiment for selected mouse samples

One mouse from each of the PBS and the PFFs groups was selected, and the images were randomly reordered to avoid counting bias influenced by memory. Counting was repeated three times at 1-week intervals by an expert (T.T.), two non-experts (K.K. and M.S.), and AI (S2b Fig in S1 File). The multipoint tool in Fiji (ImageJ, Version 1.54i, National Institutes of Health, USA) [46] was used for manual counting.

### Performance metrics of AI-based counting

The corresponding images of AI-based and manual counting were opened in Fiji (ImageJ, Version 1.54j, National Institutes of Health, USA) [46]. The SIFT algorithm was used to extract feature points [14] an affine transformation was performed using a plugin, and both counting images were aligned to create a stack image. Using manual counting as a standard (for mouse samples, counting by T.T.; for marmoset and human samples, counting by K.K.), TP, FP, and FN AI counting results were determined visually on the digital stacked images, and the performance of AI-based counting was evaluated by calculating the precision, recall, and F1-score [47].

### Statistical analysis

All statistical analysis were conducted using R (version 4.4.1) [48]. All data are shown as mean ± standard deviation (SD). SD was used differently depending on the experimental context. The population standard deviation was used to assess intra-examiner variability within the same sample through repeated counting. The sample standard deviation was used to compare between different samples, such as between manual and AI-based counting methods or between disease model and control groups. For human samples, TH-positive cell density was calculated by normalizing cell counts to the area of the corresponding ROI and is expressed as cells/mm². Pearson correlation coefficient was used to analyze the correlation between AI-based and manual counting. Agreement between AI-based and manual counting was further evaluated using Bland–Altman analysis, in which the difference between the two methods (Manual − AI) was plotted against their mean. The mean difference (bias) and the 95% limits of agreement were calculated as the mean difference ± 1.96 standard deviations. Welch's $t$-test was used to compare TH-positive cell counts between the mouse PBS and PFFs groups, and TH-positive cell density in the SNc between human control and PD patients. Statistical significance was set at 5% for all analyses. Asterisks represent level of significance: *$p < 0.05$; **$p < 0.01$; ***$p < 0.001$.

### Supporting information

**S1 File. Supplementary material including supplementary methods, legends for Supplementary Figures S1–S4, and S1–S3 Tables.**
(ZIP)

### Acknowledgments

We gratefully acknowledge Rie Hikawa, Yukako Hatate, Jun Ueda, Miki Oono, Masashi Ikuno, Satomi Toda, Junichiro Ohira, Yuji Narumiya, Jiarui Chang for technical assistance and Masateru Taniguchi for technical advice.

### Author contributions

**Conceptualization:** Ken Kunugitani, Masanori Sawamura.

**Data curation:** Ken Kunugitani, Masanori Sawamura.

**Formal analysis:** Ken Kunugitani, Masanori Sawamura.

**Investigation:** Ken Kunugitani, Masanori Sawamura, Tomoyuki Taguchi, Tetsuya Hirato, Norihito Uemura, Takashi Ayaki, Etsuro Nakanishi, Hodaka Yamakado, Tomoyuki Ishimoto, Hirotaka Onoe, Tadashi Isa, Riki Matsumoto, Ryosuke Takahashi.

**Methodology:** Ken Kunugitani, Masanori Sawamura, Tomoyuki Taguchi, Tetsuya Hirato.

**Project administration:** Masanori Sawamura.

**Resources:** Masanori Sawamura, Norihito Uemura, Takashi Ayaki, Tadashi Isa, Riki Matsumoto, Ryosuke Takahashi.

**Validation:** Ken Kunugitani, Masanori Sawamura.

**Visualization:** Ken Kunugitani, Masanori Sawamura.

**Writing – original draft:** Ken Kunugitani, Masanori Sawamura.

**Writing – review & editing:** Ken Kunugitani, Masanori Sawamura, Tomoyuki Taguchi, Tetsuya Hirato, Norihito Uemura, Takashi Ayaki, Etsuro Nakanishi, Hodaka Yamakado, Tomoyuki Ishimoto, Hirotaka Onoe, Tadashi Isa, Riki Matsumoto, Ryosuke Takahashi.

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
