## [Decision Letter · Decision Letter 0]

4 Nov 2025

PONE-D-25-53041AI-assisted quantification of dopaminergic neurons in Parkinson’s disease modelsPLOS ONE

Dear Dr. Sawamura,

Thank you for submitting your manuscript to PLOS ONE. After careful consideration, we feel that it has merit but does not fully meet PLOS ONE’s publication criteria as it currently stands. Therefore, we invite you to submit a revised version of the manuscript that addresses the points raised during the review process. In particular, we encourage you to strengthen the analysis to ensure the necessary rigour and robustness, which will help to substantiate the conclusions drawn and enhance the overall credibility of the study.

We look forward to receiving your revised manuscript.

Kind regards,

Míriam R. García

Academic Editor

PLOS ONE

Journal Requirements:

Reviewer's Responses to Questions

**Comments to the Author**

1. Is the manuscript technically sound, and do the data support the conclusions?

Reviewer #1: Yes

Reviewer #2: Partly

2. Has the statistical analysis been performed appropriately and rigorously?

Reviewer #1: Yes

Reviewer #2: No

3. Have the authors made all data underlying the findings in their manuscript fully available?

Reviewer #1: Yes

Reviewer #2: Yes

4. Is the manuscript presented in an intelligible fashion and written in standard English?

Reviewer #1: Yes

Reviewer #2: Yes

5. Review Comments to the Author

Reviewer #1: This manuscript addresses an important issue in Parkinson’s disease research: the reliable quantification of dopaminergic neurons in the substantia nigra pars compacta. The authors demonstrate that a commercially available CNN-based artificial intelligence system can quantify dopaminergic neurons with comparable accuracy to manual counting. Notably, the validation was carried out not only in mouse models but also in marmoset PD models and human PD brain samples, which is a significant strength and adds translational value to the work.

While the concept is interesting, several methodological and interpretational concerns should be clarified before the manuscript can be considered for publication.

1．Training images appear to include repeated use of the same specimens. In Supplementary Figures, it appears that for mouse and human samples, the same images (e.g., =3 or N=4) were used multiple times as teacher images. Could the authors clarify why the same specimens appear repeatedly? Clarifying these points is important because repetitive use of identical or highly similar images in a small dataset can lead to model overfitting, and therefore the reproducibility of the AI-based quantification method may be overestimated. In addition, how many individual dopaminergic neurons were used as ground-truth labels for training?

2. Because TH-positive neuronal density can vary considerably along the rostro-caudal axis of the SNc, accurate quantification of the total number of dopaminergic neurons in the SNc generally requires unbiased stereological methods, or at least a comparison with the contralateral (non-lesioned) side to correct for section-to-section variability. In this study, it is not clearly described how the authors dealt with this issue.

3. The authors state that AI-based counting was affected by staining quality, particularly in samples with weak TH staining. However, it is unclear whether such low-intensity images were included in the training set. Were teacher images selected to include a wide range of staining intensities, including weakly stained sections? Clarifying whether the training dataset captured realistic staining variability is critical for interpreting the robustness of the AI model in routine histopathological settings.

4. In the analysis of human specimens, neither AI-based nor manual counting detected a significant reduction in dopaminergic neurons in PD patients compared with healthy controls. Given that substantial neuronal loss in the SNc is a hallmark of PD pathology, this result requires further explanation. What is the authors’ interpretation of the absence of statistical significance?

Could this be due to sampling bias related to the anatomical level of the analyzed sections? As mentioned earlier, SNc neuron density varies across rostro–caudal levels. Given that the AI and manual methods both failed to show a difference, the issue may not be the counting algorithm itself, but rather the criteria used for section selection.

5. The authors cite three prior studies that applied AI approaches to quantify dopaminergic neurons. However, to my knowledge, there is at least one additional report describing similar deep-learning–based approaches for TH-positive neuron detection (Chen et al. A user-friendly deep learning approach for robust dopaminergic neuron detection. Neurosci Lett. 2024, 836:137871. doi:10.1016/j.neulet.2024.137871.). The manuscript states that the present method performs with comparable inference accuracy, but it is not clearly discussed how this method differs from, or improves upon, those existing approaches.　What specific advantages does the proposed method (TruAI) offer over previously published models?

Reviewer #2: The authors present an interesting manuscript that highlights the use of the commercial AI-assisted cell counting software, TruAI, developed by Olympus, in dopaminergic neuron counting. The study investigates the utility of this software across three use cases—Parkinson’s disease (PD) animal models with PFF injections, marmoset models with PFF injections, and human PD samples. While the findings provide valuable insights for the broader scientific community, the study would benefit from further rigor, expanded analysis, and coherence to strengthen its conclusions. Below are specific points for consideration:

1. Comments on the Mouse PFF Model:

PFF Injection Duration:

The manuscript does not clearly state the duration of PFF injections. This information is critical to understanding the timeline for induced pathology and assessing the reliability of the reported tyrosine hydroxylase (TH) loss. Please provide explicit details about the injection schedule and post-injection timepoints.

Phospho-synuclein Staining:

To confirm the presence of synuclein pathology caused by the PFF injections, phospho-synuclein staining should be performed and shown in the results. Without such confirmation, it is difficult to ascertain that the observed changes in TH cells are the result of α-synuclein pathology.

Sampling and Section Analysis:

The authors use 8 μm thick tissue sections and analyze only one section per mouse (10 sections from 10 mice). This limited sampling is insufficient for assessing the substantia nigra (SN), where TH+ neurons may vary greatly from section to section due to technical artifacts and the topology of the SN.

Although the authors mention that they captured images from many additional sections, they did not analyze these. This represents a missed opportunity to provide a more comprehensive analysis.

To justify claiming TH loss in the PFF mouse model, the authors need to evaluate the entire SN or at least perform a more systematic analysis across different rostro-caudal levels. As it stands, the authors can only claim to have counted TH cells in a limited number of sections.

Discrepancies in Sample Details:

There are inconsistencies between the sample details in the Methods section and the data presented in figures. For instance, the Methods state “10 mice (4 PBS and 6 PFFs), 5 marmosets, and 6 human (1 PD and 5 controls),” but figures show a higher number of data points than described. Please clarify and align the information throughout the manuscript.

2. Technical Questions on the Software and Methodology:

Effect of Signal Contrast:

The authors state that signal contrast impacts the accuracy of AI-based detection. Please provide specific details on:

How signal contrast was manipulated.

The magnitude of changes in contrast that led to failure in detection.

The percentage of counting errors introduced by differences in contrast.

Experience of the Testers:

Please elaborate on the qualifications and experience of the “experienced testers” who were responsible for validating the AI-based counts. Information about their expertise in histology, microscopy, or AI-assisted analysis will help gauge the reliability of their findings.

Segmentation of the Substantia Nigra (SNpc):

The images provided show a diffuse segmentation of the SNpc by the AI algorithm. How was the manual segmentation of the SNpc aligned with the AI algorithm’s segmentation? If there were discrepancies between the two methods, how were these addressed, and how do they impact the reported cell counts?

Handling of Overlapping/Dense TH+ Cells:

The TH counting algorithm seems to rely on the nuclei of individual TH cells for detection. How does the software resolve overlapping or dense TH+ cells, which are common in the SNpc and often introduce challenges in automated analysis? The representative images provided do not appear to depict such scenarios. Please discuss and provide evidence that TruAI can handle these situations, or clarify the limitations of the software in this regard.

3. Clarifications on Results and References:

Line 166:

The statement regarding a “38% decrease in 2 months” is unclear. Does this pertain to 2 months post-PFF injection, or is it referring to 2-month-old animals? Please clarify and provide appropriate citations to support this claim.

Additional Literature Review:

The literature review could be expanded to provide a more comprehensive discussion of related studies on dopaminergic neuron analysis and AI segmentation. Papers by Haghighi et al. and Barzekar et al. (2023) may provide useful comparisons and insights. Consider incorporating relevant findings from these studies into Table 3 and elsewhere in the discussion.

Training Dataset and Parameters:

Since the study aims to showcase the application of TruAI for dopaminergic neuron counting, it would be valuable to provide the training dataset and parameters used by the algorithm. This transparency will enable the broader scientific community to replicate or adapt these methods for similar applications.

4. Limitations of the Study and Overstated Conclusions:

Marmoset Model Data:

The conclusions drawn for the marmoset model appear to be accurate and appropriately supported by the data. However, the analyses of mouse and human samples do not align with the same level of rigor and should be revised accordingly. Please ensure that conclusions are aligned with the data presented.

TH Cell Counting vs. TH Loss:

The data in the manuscript convincingly demonstrate the ability of TruAI to count TH cells in different species. However, the authors’ claim that they have demonstrated the ability of the software to examine TH cell loss in the PD samples (animal models and human tissues) is not sufficiently supported. This conclusion requires additional evidence and analysis.

Human PD Sample Size:

The study analyzes only one human PD sample as per methods. With an n=1, it is not statistically sound to make broader claims about the accuracy of the AI’s detection of TH cell loss in human PD tissues. Please explain how statistical analysis (if any) was performed and provide cautionary notes about these limitations in the manuscript.

Data Normalization for Statistical Analysis:

Were the data normalized prior to statistical analysis? If so, describe the normalization process and how the data were represented in the analysis. This information is important for reproducibility and transparency.

Null Hypothesis and AI-Based Analysis:

For the AI-based analyses, what was the null hypothesis? Please explicitly discuss the hypothesis tested and the conclusions drawn based on the results.

The title can be improved and a more suitable title would be something like “Evaluation of a Commercial AI-Assisted Cell Counting Software TruAI for Dopaminergic Neuron Counting across species”

In summary, while the study demonstrates potential applications of TruAI, improvements in methodology, additional analyses, and consistency in reporting are necessary to fully support the conclusions. This will enhance the impact and utility of the work for the scientific community

6. PLOS authors have the option to publish the peer review history of their article (what does this mean?). If published, this will include your full peer review and any attached files.

Reviewer #1: **Yes:**Yasuhiko Izumi

Reviewer #2: No

---

## [Author Response · Author response to Decision Letter 1]

3 Feb 2026

We sincerely thank you and the reviewers for your thoughtful and constructive comments. The manuscript has benefited greatly from these valuable suggestions.

We have carefully revised the manuscript in accordance with all reviewer comments. Please, find the uploaded documents.

---

## [Decision Letter · Decision Letter 1]

16 Feb 2026

PONE-D-25-53041R1Evaluation of a Commercial AI-Assisted Cell Counting Software for Dopaminergic Neurons across SpeciesPLOS One

Dear Dr. Sawamura,

Thank you for submitting your manuscript to PLOS ONE. After careful consideration, we believe that it has merit, although it does not yet fully meet PLOS ONE’s publication criteria in its current form. We therefore invite you to submit a revised version of the manuscript that addresses the minor points raised during the review process.

We look forward to receiving your revised manuscript.

Kind regards,

Míriam R. García

Academic Editor

PLOS One

Journal Requirements:

Reviewers' comments:

Reviewer's Responses to Questions

**Comments to the Author**

1. If the authors have adequately addressed your comments raised in a previous round of review and you feel that this manuscript is now acceptable for publication, you may indicate that here to bypass the “Comments to the Author” section, enter your conflict of interest statement in the “Confidential to Editor” section, and submit your "Accept" recommendation.

Reviewer #2: All comments have been addressed

2. Is the manuscript technically sound, and do the data support the conclusions?

Reviewer #2: Yes

3. Has the statistical analysis been performed appropriately and rigorously?

Reviewer #2: Yes

4. Have the authors made all data underlying the findings in their manuscript fully available?

Reviewer #2: Yes

5. Is the manuscript presented in an intelligible fashion and written in standard English?

Reviewer #2: Yes

6. Review Comments to the Author

Reviewer #2: Thank you for thoroughly addressing all the comments. I understand that it is painful to address the reviewers comments but I hope this will make the manuscript stronger and will receive credits in the long run.

I have some very minor comments to better understand,

1.Were reference images also incorporated in counting the cells, if not, why was that route chosen? Better to have counts from all images considering there is limited images in the study which has been highlighted.

2. Under "AI-based counting of TH-positive cells in mouse, marmoset, and human samples" it still says 6 human samples but "Human Samples" in methods says 9. Please cross check.

3. I didn't see adding Cellpose model as a reference and addressing why this is better than that considering it is the state of art model for cell counting. Additionally, can the authors please add how the model can be made better, for e.g incorporating automated segmentation models for SN as has been shown in some recent manuscripts. That might provide the company to make their workflow and software better for future users.

7. PLOS authors have the option to publish the peer review history of their article (what does this mean?). If published, this will include your full peer review and any attached files.

Reviewer #2: No

---

## [Author Response · Author response to Decision Letter 2]

21 Feb 2026

We are pleased to submit a further revised version of our manuscript, entitled “Evaluation of a Commercial AI-Assisted Cell Counting Software for Dopaminergic Neurons across Species,” to PLOS One. The manuscript ID is PONE-D-25-53041.

We sincerely thank you and the reviewers for your careful evaluation of our manuscript and for the additional insightful comments. These comments have been invaluable in further improving the clarity and rigor of our work.

We have carefully addressed all reviewer comments and revised the manuscript accordingly. Detailed responses to each comment are uploaded. Please, check the attached files.

---

## [Editor Report · Decision Letter 2]

23 Feb 2026

Evaluation of a Commercial AI-Assisted Cell Counting Software for Dopaminergic Neurons across Species

PONE-D-25-53041R2

Dear Dr. Sawamura,

We’re pleased to inform you that your manuscript has been judged scientifically suitable for publication and will be formally accepted for publication once it meets all outstanding technical requirements.

Kind regards,

Míriam R. García

Academic Editor

PLOS One

---

## [Editor Report · Acceptance letter]

PONE-D-25-53041R2

PLOS One

Dear Dr. Sawamura,

I'm pleased to inform you that your manuscript has been deemed suitable for publication in PLOS One. Congratulations! Your manuscript is now being handed over to our production team.

Kind regards,

on behalf of

Dr. Míriam R. García

Academic Editor

PLOS One